# Optical Imaging Deformation Inspection and Quality Level Determination of Multifocal Glasses

**DOI:** 10.3390/s23094497

**Published:** 2023-05-05

**Authors:** Hong-Dar Lin, Tung-Hsin Lee, Chou-Hsien Lin, Hsin-Chieh Wu

**Affiliations:** 1Department of Industrial Engineering and Management, Chaoyang University of Technology, Taichung 413310, Taiwan; hdlin@cyut.edu.tw (H.-D.L.); tung@insighteyes.com (T.-H.L.); 2Department of Civil, Architectural, and Environmental Engineering, The University of Texas at Austin, Austin, TX 78712-0273, USA; chslin@utexas.edu

**Keywords:** multifocal glasses, deformation inspection, quality level determination, slight deviation control scheme, fuzzy theory, genetic algorithm

## Abstract

Multifocal glasses are a new type of lens that can fit both nearsighted and farsighted vision on the same lens. This property allows the glass to have various curvatures in distinct regions within the glass during the grinding process. However, when the curvature varies irregularly, the glass is prone to optical deformation during imaging. Most of the previous studies on imaging deformation focus on the deformation correction of optical lenses. Consequently, this research uses an automatic deformation defect detection system for multifocal glasses to replace professional assessors. To quantify the grade of deformation of curved multifocal glasses, we first digitally imaged a pattern of concentric circles through a test glass to generate an imaged image of the glass. Second, we preprocess the image to enhance the clarity of the concentric circles’ appearance. A centroid-radius model is used to represent the form variation properties of every circle in the processed image. Third, the deviation of the centroid radius for detecting deformation defects is found by a slight deviation control scheme, and we gain a difference image indicating the detected deformed regions after comparing it with the norm pattern. Fourth, based on the deformation measure and occurrence location of multifocal glasses, we build fuzzy membership functions and inference regulations to quantify the deformation’s severity. Finally, a mixed model incorporating a network-based fuzzy inference and a genetic algorithm is applied to determine a quality grade for the deformation severity of detected defects. Testing outcomes show that the proposed methods attain a 94% accuracy rate of the quality levels for deformation severity, an 81% recall rate of deformation defects, and an 11% false positive rate for multifocal glass detection. This research contributes solutions to the problems of imaging deformation inspection and provides computer-aided systems for determining quality levels that meet the demands of inspection and quality control.

## 1. Introduction

Curved components can often meet a wide range of structural requirements because they allow greater geometric freedom in design. They can establish the general look and feel of the design [1]. Components with curved surfaces are popular in engineering practices such as automotive, aerospace, optics, etc. A glass is a piece of a lens, plastic, or other transparent material that is curved on one or both sides; these curves bend the light passing through the lens. Such optical devices tend to converge and diverge light beams through refraction to produce optical images. However, components with curved surfaces often need to be inspected, which can lead to complications that reduce the sensitivity of defect detection and increase the chance of false alarms [2,3].

Multifocal glasses are a new type of lens that can fit both nearsighted and farsighted vision on the same lens. This property allows the glass to have various curvatures in distinct regions within the glass during the grinding process. However, the disadvantage of multifocal glasses is that when the curvature changes abnormally, the glasses are prone to optical deformation during imaging. Because the glasses are utilized to directly envelop the eyes of the user, the deformation of the glass will cause imaging errors, which will bring trouble or even danger to the daily activities of the user. For example, dizziness may occur when going downstairs, or the inability to correctly decide the location between steps may be hazardous for the person. Figure 1 shows diagrammatic sketches of the imaging deformations of the stairs at close range using defective and normal multifocal glasses, respectively.

Progressive multifocal glasses allow people with myopia and presbyopia to easily and effectively improve their vision [4,5]. The progressive multifocal glass has areas of nearsighted, farsighted, progressive, and blindness, as shown in Figure 2. The advantage of wearing progressive multifocal glasses is that when you need to see farther places (scenery, buildings, etc.), you can use the area of the top half of the lens; when you need to see a closer place (mobile phone, book, etc.), you can use the bottom half area of the lens; the middle region is the progressive area for adjustment. The disadvantage of this lens is that there will be blind spots in the lower right and lower left. Frequent deformations in the blind areas also need attention since these deformations will explicitly influence the imaging quality of objects viewed by users.

Although the blind spot location is usually not larger than in the other areas, the use of this area is very important and requires special inspection. At present, the degree of deformation of the blind area is inspected by inspectors operating manual instruments. Since imaging deformation has no regular form and clear borders, it is frequently difficult to identify and quantify, particularly on curved multifocal glasses. Furthermore, outwardly curved glasses tend to hinder the identification of deformation defects in multifocal glasses due to their high transmittance and reflectivity. This study develops an optical inspection system to quickly inspect and classify deformation defects in multifocal glasses to replace professional assessors.

The rest of the article is composed as follows. First, we review the articles on the current techniques used by visual inspection systems for transparent products. Second, we describe the proposed image procedures to detect imaging deformation defects and determine the quality level of multifocal glasses. Third, we conduct tests and assess the performance of the suggested approach and traditional techniques. Finally, we conclude the contributions and indicate further directions.

## 2. Literature Review

Automated visual inspection (AVI) is a crucial step in quality assessment for production processes as it assures high quality and enhances productivity by rigorously examining and evaluating all products in an industrial process [6,7]. The AVI system integrates computer vision and machine learning techniques and has been widely applied in various industries [8]. To improve the quality of glass products, many researchers have developed automated optical inspection devices using advanced image processing methods to detect the shape and surface of glass products and analyze their optical properties. These studies focus on detecting surface defects in industrial precision glass-related products, such as: applying precise band Gaussian filtering based on discrete cosine space to detect the appearance flaws of capacitive touch screens with directional textures [9], proposing an optical detection method for aspheric glasses in semiconductor sensor modules [10], combining Hilbert–Huang transform with random forest model to locate the flaw positions on the front or back side of the screens [11], and using convex hull algorithm after Fourier filtering to implement car mirror detection structure [12]. These vision inspection systems mainly target surface defects of glass-related industrial products.

Image deformation caused by perspective requires image correction before further image processing [13]. In previous studies on deformation detection and correction methods, Mantel et al. [14] proposed a technique for identifying perspective deformation in photoluminescence images of photovoltaic screens, and Cutolo et al. [15] presented fast methods for calibrating see-through head-mounted displays using calibration camera. Apparently, most studies on perspective-related deformations have focused on deformation correction for optical glass [16,17].

Transmission deformation refers to the image quality degradation of an observed object due to refractive media. Previous studies on deformation detection via the transmission of industrial components include Dixon [18] who designed a system to measure optical deformation in aircraft transparencies using digital imaging and a decision tree-based classifier. Youngquist et al. [19] introduced a new method of interpreting transmitted deformations and enabled the use of phase-shifting interferometers to estimate deformations in large optical windows. Chiu et al. [20] constructed an optical system based on small drift control charts to identify deformation defects in curved car mirrors. Gerton et al. [21] analytically investigated the deformation patterns of Ronchi meshes to assess the impact of deformation on eyewear products. Lin et al. [22] employed the Hough transform voting scheme to inspect deformation defects for see-through glass products.

Glasses are wearable products that are closely related to human daily activities and require strict testing and verification. If the eyeglass lenses have excessive defects, the user’s safety will be compromised. Wang et al. [23] performed a symmetric energy analysis on different color spaces to evaluate the coating quality of glass. Yao et al. [24] employed low-angle LED (light-emitting diode) illumination and image normalization techniques for computer vision and lens categorization to identify glass surface defects. Karangwa et al. [25] developed a visual inspection platform integrating deep learning models and semantic segmentation to detect and classify the visual defects of optical glass. Lin et al. [26] designed an optical inspection system based on computer vision for various optical components, such as camera lenses, glasses, and other optical devices. Lin [27] proposed a method based on adaptive vision, combining wavelet feature extraction and support vector machine classification to classify lens images and determine the grade of eyeglasses.

Currently, most optical inspection systems for transparent glass products mainly detect surface defects and do not identify imaging deformation defects. The imaging deformation defects in curved glass surfaces are highly transmissive and reflective and are difficult to accurately detect [20]. Most of the related studies on perspective deformation focus on the deformation correction of optical glasses. At present, few works have applied optical inspection systems to detect imaging deformation defects in eyeglasses. Therefore, we propose an optical inspection system based on a slight deviation control scheme to detect imaging deformation defects in multifocal glasses. With proper parameter settings and robustness analysis, the method can recognize not only severe deformation defects but also slight deformation defects.

## 3. Research Method

This study presents an optical inspection system with familiar norm patterns of image capture and employs a slight deviation control scheme to check for deformation defects, and a mixed intelligent model incorporating a genetic algorithm and a network-based fuzzy inference system to decide the grade of deformation severity of the multifocal glasses. To measure the grade of deformation of a multifocal glass, we first digitally imaged a pattern of concentric circles through a test glass to generate an imaged image of the glass deformation. This deformation image is examined to investigate the deformation’s presence and the defects’ location. Secondly, the imaged image is preprocessed to enhance the clarity of the concentric circles’ appearance. A centroid-radius model is adopted to represent the form variation properties of every concentric circle in the processed image. Thirdly, by looking for small changes in characteristic distance deviations to detect deformation defects through a slight deviation control scheme, a difference image showing the detected deformation defects can be obtained. Fourthly, according to the deformation measurements and locations that occurred during the training phase, the deformation’s fuzzy membership functions and inference rule sets are established. Finally, a mixed model incorporating a genetic algorithm and a fuzzy inference model is adopted to judge the grade in the deformation severity of the detected deformation defects. Figure 3 shows the workflow of the stages of the proposed approach.

### 3.1. Image Capture and Image Preprocessing

The adopted test samples are 6.0 mm thick and 48.4 mm in diameter and are arbitrarily drawn from the production line of a multifocal glass producer. In order to capture digital imaging images of norm patterns by testing samples to create imaging deformation maps of samples, this work suggests an imaging acquisition device using a concentric circular pattern for imaging image extraction. Figure 4 illustrates the arrangement of the image acquisition apparatus to capture the test images of multifocal glasses. The test sample is inserted horizontally into the custom-made fixture in front of the norm pattern. A norm pattern of essentially concentric circles is placed on the foundation of the stand. A camera with a mount is applied to capture images from the sight transferred through the test glass on a pattern of concentric circles. To capture a digital image of a norm pattern with appropriate brightness, the light source control of the surroundings in which the image is acquired is also important.

Figure 5a,b shows two images captured from transmission imaging of the concentric circular pattern through a normal multifocal glass and a defective glass, respectively. The flawed image has noticeable deformation in the upper right area. The acquired images are preprocessed in a number of stages to enhance the clarity of the appearance of objects on light-transmitting glasses. In order to quantify the degree of deformation of the acquired pattern images, Figure 5c depicts the binarized and thinned images of defect samples by applying the Otsu method [28] and thinning algorithm [29] to perform segmentation and thinning operations sequentially when using a concentric circular pattern. With these two methods, most concentric circles are separated from the background and thinned to become binary and 1-pixel width images. The results show that moderately deformed defects on transparent glass surfaces are correctly segmented in binary images, regardless of small differences in deformation.

### 3.2. Feature Representation

When using coordinates for image feature processing, some problems arise. When the image is translated, zoomed, or rotated, the judgment result will be erroneous due to the coordinate changes. Hence, it should be depicted by geometric properties. We use the centroid-radius model [30] to represent the geometrical properties of every circle in the image by calculating the distances from the edge points to the centroid. The coordinates of the edge points are transformed into vectors of distance properties by the Euclidean metric. The reason for using Euclidean distance is that it is invariant to translation, scaling, and rotation.

We use the commonly used concentric circular pattern, which consists of eight concentric circles. The centroid radius $r_{s}^{u}$ is the Euclidean distance calculated from the centroid O(x_0_, y_0_) and the *s*-th boundary point (x_s,u_, y_s,u_). The distances to the *u*-th circle in the concentric circular pattern are:(1)$$r_{s}^{u}=\sqrt{{\left(x_{s,u}-x_{0}\right)}^{2}+{\left(y_{s,u}-y_{0}\right)}^{2}},s=1,2,3,\;\ldots$$

For this pattern of concentric circles, many centroid radii of the *u*-th concentric circle can yield a distance vector $R_{u}$, expressed as follows:
(2)$$R_{u}=\{r_{1}^{u},r_{2}^{u},r_{3}^{u},\ldots r_{s}^{u},\ldots \},u=1,2,3,\;\ldots ,8$$


Due to scale and rotation invariance, we normalize the distance vector $R_{u}$ to 0 and 1 by dividing each value by the maximum value of the distance vector to obtain a normalized distance vector $Q_{u}$:(3)$$Q_{u}=\left\{q_{1}^{u},q_{2}^{u},q_{3}^{u},\ldots ,q_{s}^{u},\ldots \right\},\;\mathrm{where}\;q_{s}^{u}=r_{s}^{u}/\max(r_{s}^{u})$$

When calculating the distances of centroid radii from the *u*-th circle to the circle’s centroid and normalizing it to a vector $Q_{u}$, all points and Euclidean distances from the feature vector can be plotted, as shown in Figure 6a. Figure 6b indicates that the more the normalized distance values are, the farther they are from the center position, and the potential deformation in these concentric circle regions is more obvious.

### 3.3. Deformation Detection by a Slight Deviation Control Scheme

The distance feature vectors of all complete concentric circles in a test image are contrasted with those of the defect-free image, and the distance deviations of the respective edge points are measured to locate potential deformations in the test image. To detect slight deviations in distance variations, we propose a slight deviation control scheme, the exponentially weighted moving average (EWMA) scheme, which is usually applied in statistical process control [31,32]. We apply the EWMA scheme to find slight changes in distance deviations to detect deformation defects.

The EWMA scheme is also a good option to detect slight drifts [33,34]. The exponentially weighted moving average $Z_{s}$ with the *s*-th sample point is defined as:(4)$$Z_{s}=\lambda \;q_{s}+\left(1-\lambda \right)\;Z_{s-1}$$
where the initial value of $Z_{s}$ is the target value $Z_{0}\;$=$\;\mu_{0}$ and the λ is the named constant weight located in the space 0 < λ ≦ 1. The value of parameter λ in the space 0.05~0.25 is suitable for the detection of slight deviations in practical application. A recommended rule is to adopt smaller λ values to detect slight variations. The control limit of the upper bound (*UCL*) and the control limit of lower bound (*LCL*) for the EWMA scheme are expressed as [33]:(5)$$UCL_{s}=\mu_{0}+L\sigma \sqrt{\frac{\lambda }{2-\lambda }\left[1-{\left(1-\lambda \right)}^{2s}\right]}$$
(6)$$LCL_{s}=\mu_{0}-L\sigma \sqrt{\frac{\lambda }{2-\lambda }\left[1-{\left(1-\lambda \right)}^{2s}\right]}$$

The parameter designs of the chart are the multiple of the standard deviation σ applied in the control limits (*L*) and the value of constant weight $\left(\lambda \right)$. The capability of the EWMA scheme is roughly comparable to that of the CUSUM scheme, and in some respects, it is simpler to establish the model and manipulate the schema [35,36]. Figure 7a is the output of the fifth circle in the concentric circular pattern of a test image performed by the EWMA scheme. Figure 7b shows when EWMA is used to detect the deformation defect regions; the boundary point range of the defect position will thus be clearer.

### 3.4. Quality Level Determination of Deformation Severity by the Fuzzy Inference System

This research applies a fuzzy-related model to automatically detect changes in deformation severity [37]. The genetic algorithm-based adaptive neuro-fuzzy inference system (GA-based ANFIS) model combines genetic algorithms and adaptive-network-based fuzzy inference theory, consisting of FIS and back-propagation network (BPN). In this study, a GA-based ANFIS model is used to judge the quality grade of deformation in multifocal glasses. When the merits of these techniques are combined, the judgment accuracy of the detection system will be notably enhanced.

By contrasting the detected deformation defect image with the concentric circular pattern image, Figure 8a shows that the regions labeled by red lines are the deformation defects, and the regions labeled by white lines are the norm pattern. In addition, the points on the found defects are contrasted to the corresponding points in the concentric circular pattern, respectively. The Manhattan metric vector U denoting the measurement of the deformation amount is defined as follows:(7)$$u_{s}=\left|x_{m,s}-x_{n,s}\right|+\left|y_{m,s}-y_{n,s}\right|$$
(8)$$U=\left\{u_{1},u_{2},u_{3},\;\ldots ,u_{s},\;\ldots \right\},\;s=1,2,3,\;\ldots$$
where $x_{m,s}$ and $y_{m,s}$ are the (*x*, *y*) coordinates of the *s*-th edge point in a test image, $x_{n,s}$ and $y_{n,s}$ are the corresponding (*x*, *y*) coordinates of the *s*-th edge point in the concentric circular pattern image.

Figure 8b shows a difference image labeled according to three different deformation severities. The image is distinguished as three regions according to three distinct degrees of deformation: the red-line portion is the low allowance area, which comprises the first and second circles, named zone A; the blue-line portion is the medium allowance area, which consists of the third, fourth, and fifth circles, named zone B; the green-line portion is the high allowance area, composed of the rest circles, named zone C. The measurements of the individual deformations of the three zones are fed into the model for fuzzy inference and grade judgment.

#### 3.4.1. Fuzzy Inference System of Deformation Levels

The main purpose of the FIS model is not only to transform the measured values into fuzzy membership functions but also to set up fuzzy inference regulations and modes [38,39]. Its merit is that, while the input is fuzzy details, an appropriate corresponding value can be output through the process of establishing inference rules and defuzzification algorithms. In this work, the measures of deformation in zones A, B, and C are employed as the input items to categorize the severities of glass deformation. When performing fuzzy inference, it is first necessary to transform the feature values of the distortion variables into membership functions. We use Gaussian membership functions to set the range of feature values as follows:(9)$$Gaussian\left(u;\sigma ,c\right)=e^{-\frac{1}{2}{\left(\frac{u-c}{\sigma }\right)}^{2}}$$
where *σ* is the standard deviation and *c* is the center point of the Gaussian membership function.

Table 1 summarizes three features as the input items of the FIS model, which are the deformation degrees of the three zones, A, B, and C, respectively, and the output item is the quality grade of deformation. In the boundary settings of the fuzzy sets of the three input measures, while the input is the deformation of zone A, due to the low allowance, it is set to two levels, and the other inputs and outputs are three levels. The membership functions and fuzzy set definitions of the input items are shown in Table 2.

When the fuzzy membership functions have been set up, the fuzzy regulation base could be created based on the allowance of the deformation level of the multifocal glass. The closer the deformation defect occurs to the middle of the glass, the smaller the allowed allowance is, and the degree of deformation is severe. The closer the deformation flaw region is to the glass boundary, the higher the tolerance is allowed and is then categorized as slight on the deformation level. The fuzzy regulations are created according to the empirical regulations of experts. Three inputs, U_1_, U_2_, and U_3_, are the deformation levels of the zones A, B, and C, respectively, and the outputs are deformation levels. For example, while the deformation measure U_1_ of zone A is small (A_1_) and the deformation measure U_2_ of zone B is small (B_1_) and the deformation measure U_3_ of zone C is small (C_1_), the severity of the output Y is a slight distortion (Y_1_). A fuzzy regulation base containing eighteen regulations is created in this experiment.

We use the TSK (Takagi–Sugeno–Kang) fuzzy model [39] as an inference engine, and the model combines the application of fuzzy regulations using the IF-THEN format. The result of each regulation is a linear combination of the input factors and the constant term. The final result is a weighted average of the results of each rule. It mostly employs fuzzy regulations to describe a non-linear system. The merits of this approach are rapid computational efficiency, good cooperation with adaptive optimization techniques, and continual output values, which are very suitable for arithmetical analysis. While the deformation degrees of the three zones are input into the FIS and deduced by all the regulations, the accurate output values can be produced by means of the defuzzification procedure via applying the weighted average scheme. After the outcomes of all regulations are calculated, the last output could be determined.

#### 3.4.2. Adaptive Neuro-Fuzzy Inference System for Determining Quality Level of Deformations

ANFIS model is mostly a network-based fuzzy inference model set up by merging the concepts of fuzzy theory and neural networks [40,41]. It is assessed by repeatedly varying the values of parameters and minimizing the error functions. This work establishes a five-layer architecture diagram of the ANFIS model by means of the deformation measurements of the three zones of the five layers—input layer, regulation layer, normalization layer, inference layer, and output layer—as indicated in Figure 9. By means of the learning process of the method, the training can be performed iteratively, and the parameters will be repeatedly corrected by calculating the error values of the parameters during each training. While the training errors converge to the least values or to the number of training times and achieves the preset largest number of times, the training procedure is terminated to obtain a finer trained network-based FIS than the initial parameter setting.

#### 3.4.3. Genetic Algorithm (GA)-Based Adaptive Neuro-Fuzzy Inference System for Determining Quality Level of Deformations

Genetic algorithm mostly uses the procedure of reproduction and inheritance of organisms by means of mutation and crossover of chromosomes. Firstly, the original population of the GA is established by using the membership function parameters of the deformation measures, and the fitness values of each parameter combination are evaluated. The parameters of every parameter combination are then uniformly crossed and mutated, resulting in more diverse parameter values. The merit of this arithmetic is that it could be upgraded from a local optimal answer to a globally optimal solution by a mutation function.

When a fuzzy inference system is created, it consists of membership functions of the three zones of deformations, the regulation set, and the inference module. The GA-based ANFIS method primarily optimizes the arithmetic in two stages [42,43]. The first stage in optimization is to apply ANFIS to calculate the deviations between forecasted values and true solutions, and optimize the answers by the gradient descent scheme. However, applying the gradient descent scheme could only search for a local optimum answer. The other stage of optimization is to apply GA to assess the fitness values of parameter combinations and choose finer parameters from crossover to share information, and eventually to mutate to widen the ranges of practicable parameters. Upgrading the previously obtained local optimum answer to the global optimum answer is the goal of the second stage.

## 4. Experiments and Results

To confirm the capabilities of the proposed approach using concentric circular patterns, the performances of these recommended techniques are assessed on 550 sample images (350 training images and 200 test images) with different degrees of deformation. Each captured image has a size of 256 × 256 pixels, and each pixel contains 8 bits of gray scale. The algorithm of the realized deformation defect detection system is edited on the MATLAB application software platform, and implemented on the MATLAB R2013 version on the desktop (INTEL CORE i5-8250U 1.60 GHz, 32 GB RAM). Figure 10 shows the user interface design of the developed deformation defect detection system, showing all the processing steps using the concentric circular pattern in the multifocal glass.

In this study, experimentally detected images are compared for correctness with manually labeled images. In terms of checking deformation defects, recall, precision, and accuracy are adopted as performance evaluation metrics for the proposed models. When the above performance indices are higher, the inspection performance is better. The recall rate is the regions of correctly identified true defects (True Positives, TP) divided by the regions of correctly identified true defects (TP) plus the regions of true defects incorrectly labeled as non-defects (False Negatives, FN). It can be thought of as the fraction of true defects that are correctly identified in the set of all real defects. The precision rate is the regions of exactly identified true defects (TP) divided by the regions of correctly identified true defects (TP) plus the regions of non-defects incorrectly labeled as defects (False Positives, FP). It can be thought of as the fraction of these detected defects that are real defects. The accuracy rate is the regions of exactly identified true defects (TP) and the regions of correctly labeled true non-defects (TN) divided by the total regions of a testing instance (TP + TN + FP + FN). It represents the rate of correctly identified defects and non-defects over the total region of a testing image. If the dataset is imbalanced (both defect and non-defect classes have significantly different regions of testing images), the accuracy rate is not a good metric [44,45].

In terms of determining the deformation quality level, the accuracy rate is modified from individual deformation defects to a performance evaluation index with individual images as the basic unit. The detected deformations in an image are judged combinedly into three categories (slight, average, and severe) by the GA-based ANFIS model and they are checked that each class is correctly classified. The accuracy rate is the number of test images categorized into the exact level class divided by the total number of test images.

### 4.1. Performance Assessment of Various Line Thicknesses in the Concentric Circle Patterns

The size in pixel units of the line thickness of each circle on the concentric standard patterns influences the detection efficiency of the suggested approach for deformation defects. Smaller deformation defects will be more completely identified if an appropriate line thickness pixel size is chosen in the concentric standard pattern. We use a computer program to generate standard concentric patterns with different line thicknesses and then use these standard patterns with different line widths together with test samples to capture test images. We examine concentric circular patterns with line widths from 1 to 6 pixels in concentric circles by the suggested approach. Figure 11 shows the images acquired by the suggested method, employing patterns of concentric circles with line widths of five kinds of pixel sizes and the results of a defect sample. We find that concentric circles with a thickness of one pixel are less sensitive to the detection of deformation defects, resulting in the lowest recall rate. On the other hand, concentric circles with a greater pixel thickness are more sensitized to the detection of deformation defects and lead to more false positive alarms. Table 3 denotes that the detection outcomes of the concentric circular patterns with line widths of 2 pixels and 3 pixels are more appropriate, with a higher recall rate and a lower false positive rate and a better deformation detection performance.

### 4.2. Performance Assessment of Applying EWMA Slight Deviation Control Scheme

In order to evaluate the inspection performance of multifocal glass deformation defects, Table 4 summarizes the detection and quality level judgment outcomes of the approach suggested in this work. The EWMA slight deviation control scheme of the proposed method is evaluated based on the outcomes by expertise assessors. The average recall rate of deformation inspection across all tests performed by the EWMA scheme is 81.09%. However, the precision rate of the EWMA scheme is significantly higher at 89.06%. The proposed EWMA scheme has a high deformation recall rate and a low false positive rate. Figure 12 shows some outcomes of concentric imaging deformation inspection of the suggested method employing the EWMA slight deviation control scheme. The mean execution time for a test image with a size of 256 × 256 pixels is 0.2847 s by the EWMA scheme. Thus, the proposed EWMA scheme overcomes the difficulty of detecting deformation defects in multifocal glasses and goes beyond its capability to correctly distinguish slight deformation defects from normal areas.

To assess the performance of the classification of deformation defect severity in multifocal glasses, three classification models BPN [46], ANFIS, and GA-based ANIFS are further evaluated based on the results of expertise assessors. It can be seen from Table 4 that no matter what deformation level classification model is applied, the accuracy rate of the deformation grade of the GA-based ANFIS method is larger than those of the BPN and ANFIS models. Based on the above analysis, we find that the suggested mixed method incorporating the EWMA scheme and the GA-based ANFIS method is a superior slight-deviation detection and grade determination technique for multifocal lens imaging deformation detection and severity judgment.

### 4.3. Performance Assessment of Using Distinct Norm Patterns in Deformation Detection by the Suggested Method

Using the method based on the Hough transform [22], two conventional norm patterns, a checkered pattern, and a dot pattern, were applied to detect deformations to differentiate the results of deformation defect detection. Because we adopt the three norm patterns to create consistent deformation defect images by choosing the same deformed locations and deformation degrees, the deformation distinctions among the three norm patterns can be contrasted more precisely. To show the impact of deformation detection on the consistent deformation images, Figure 13 shows some detection results of the Hough transform-based method, proposed method, and inspector for deformation defects using a checkered pattern, a dot pattern, and a concentric circular pattern, respectively. The Hough transform-based method with the checkered pattern makes many wrong identifications not only in missing alarms but also in false positives, and the same method with the dot pattern also results in some wrong identifications in missing alarms and false positives on the deformation defect inspection of multifocal glasses. The suggested approach with the concentric circular pattern can check most of the deformation defects and make less false identifications. Table 5 sums up the results of the imaging deformation inspection by the Hough transform-based methods and the suggested approach using the three standard modes. This demonstrates that the suggested approach with the concentric circular pattern outperforms the current techniques using the checker pattern and dot pattern in the deformation defect detection of multifocal glass images.

### 4.4. Robustness Tests on Changing the Brightness of the Image Illumination for Deformation Detection Results by the Suggested Approach

This study uses different parameter settings of the capture-related devices to take various test image sets and then select the test image set with the best defect detection effect. The parameter settings of this selected test image set are used as the standard for subsequent image acquisition. During the image acquisition process, the brightness of the acquired image is easily affected by the intensity of ambient light, which in turn affects the detection results. We investigate whether the detection results of the proposed method are susceptible to certain changes in image brightness to test the robustness of the method. In this study, the EWMA control scheme is applied to detect deformation defects, and the GA-based ANFIS method is adopted to categorize the severity of deformation defects. The performance evaluation results using different brightness variations are shown in Table 6, and the PR (precision–recall) chart [47,48] showing the detection performance variation trend is plotted in Figure 14. It can be seen that when the brightness of the image becomes brighter or darker by more than one standard deviation, the detection recall rate decreases significantly, and the imprecision rate also increases significantly. Figure 15 shows the local deformation detection results of the proposed method to systematically vary the image illumination brightness. Although some deformed regions are missing, most of them are detected and the overall detection rate is still better when the original brightness is applied. The proposed method is moderately sensitive to changes in light intensity. The results show that deformation defects in most multifocal glass images are accurately identified in the resulting images despite slight and moderate illumination changes.

The proposed concentric circular pattern outperforms the other two norm patterns in detecting slight to average deformations and small to medium area deformations. Therefore, the use of concentric circular patterns is more suitable for detecting deformation defects with less deformation in multifocal glasses. The main merit of this research method is to use the centroid radius descriptor to understand the deformation state of each edge point and use the EWMA control scheme to detect small deformations. In addition, using the GA-based ANFIS classifier model, the parameters that only converge in the local domain are extended to the global domain, which improves the classification effect.

Since the proposed method is mainly established on extracting features from geometric properties for deformation detection, it is moderately sensitive to changes in illumination intensity. If the brightness variation range is within (μ ± 1σ) as a whole, it will have little effect on the detection results of this study. However, large changes in illumination can significantly increase grayscale variation, which in turn can significantly affect defect detection. To conquer the restrictions of the proposed method, it is suggested to update the statistics (mean and standard deviation) of the intensity in the training samples when the illumination changes significantly.

## 5. Conclusions

This research presents a mixed approach constructed using computer vision and fuzzy theory techniques to detect deformation defects and decide the deformation level of multifocal glasses. It investigates the detection of imaging deformation defects in multifocal glass images and the classification of deformation severity. In this study, a vision system using concentric circular patterns for imaging is first developed to obtain test images showing imaging deformed areas and binarize and refine the circle edges in the image. If the boundary point-to-centroid distance value of the concentric circle goes beyond the upper or lower limit of the suggested EWMA scheme, it indicates that there is a deformation defect in this boundary point area. Then, through comparing the discovered defect image, the norm pattern is used to measure the amount of deformation. By partitioning the probable locations of defects into three zones of slight, average, and severe deformation, we summarize the individual deformation measures of the three zones. Finally, a GA-based ANFIS model is suggested to categorize the severity of multifocal glass deformation. The suggested approach is effective and efficient in detecting deformation defects and classifying the severity of deformed regions on multifocal glass images. Testing outcomes show that the proposed methods attain a 94% accuracy rate of deformation severity quality grades, an 81% recall rate of deformation defects, and an 11% false positive rate in multifocal glass deformation detection. Further studies can extend the proposed method to the problem of imaging deformation defect inspection of curved glass-related products, for example, the distortion detection of automobile windshields and the deformation detection of automobile rearview mirrors.

## Figures and Tables

**Figure 1 sensors-23-04497-f001:**
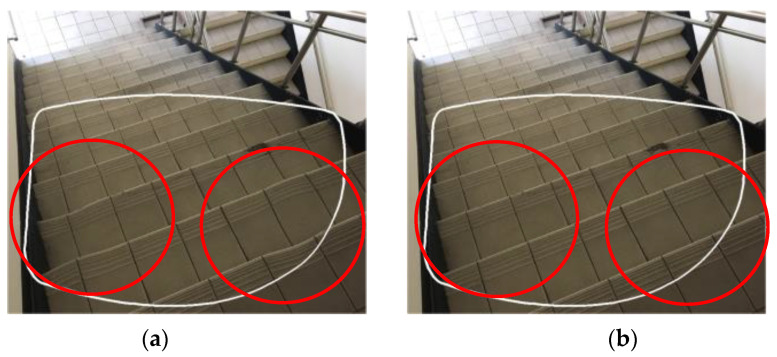
Diagrammatic sketches of the imaging deformation of the stairs at close range using multifocal glasses: (**a**) defective glasses, (**b**) normal glasses. (The range of the white frame is the field of view of the eyes wearing glasses, and the range of the red circle is the place where the imaging area is deformed).

**Figure 2 sensors-23-04497-f002:**
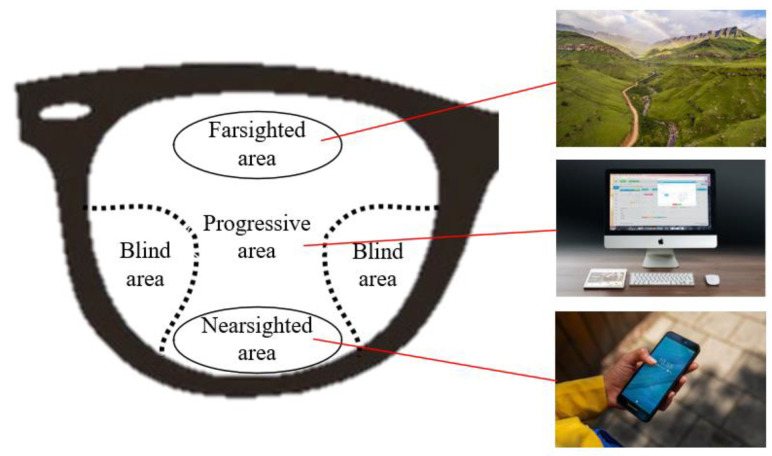
Functional block diagram of a progressive multifocal glass.

**Figure 3 sensors-23-04497-f003:**
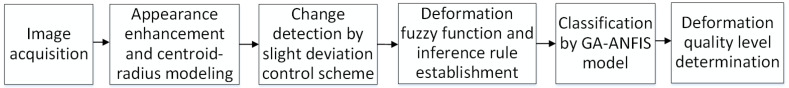
The workflow of the stages of the proposed approach.

**Figure 4 sensors-23-04497-f004:**
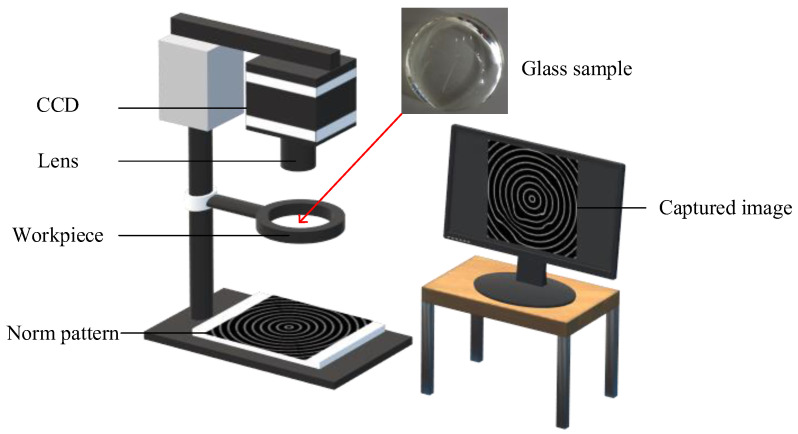
The proposed image acquisition system using the concentric circular pattern for image capture.

**Figure 5 sensors-23-04497-f005:**
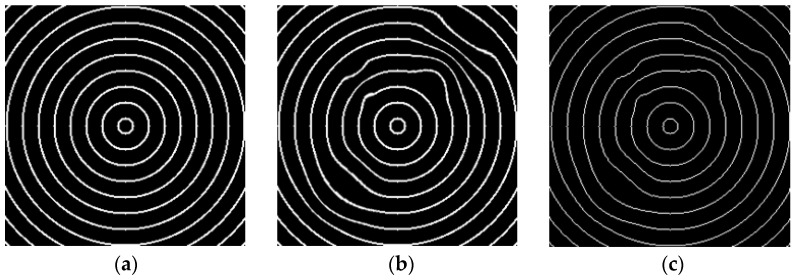
Two images captured from transmission imaging of the concentric circular pattern by a multifocal glass: (**a**) normal sample; (**b**) defective sample; (**c**) binarized and thinned image of the defective sample.

**Figure 6 sensors-23-04497-f006:**
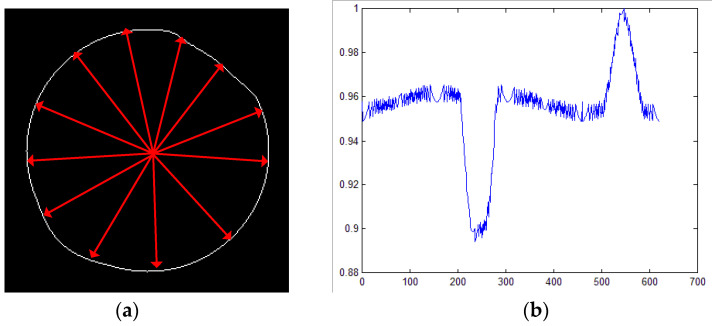
The distance values of edge points to the centroid in a concentric circle: (**a**) Euclidean distance diagram; (**b**) the edge points and corresponding normalized distance values.

**Figure 7 sensors-23-04497-f007:**
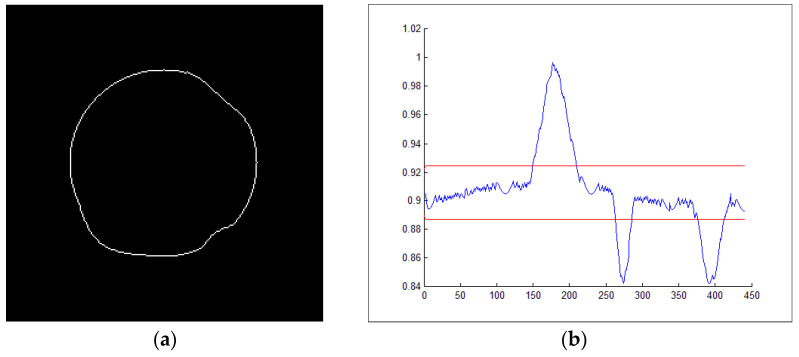
The outputs of the fifth circle in the concentric circular pattern of a test image performed by the EWMA scheme: (**a**) edge points of the fifth circle; (**b**) EWMA control chart.

**Figure 8 sensors-23-04497-f008:**
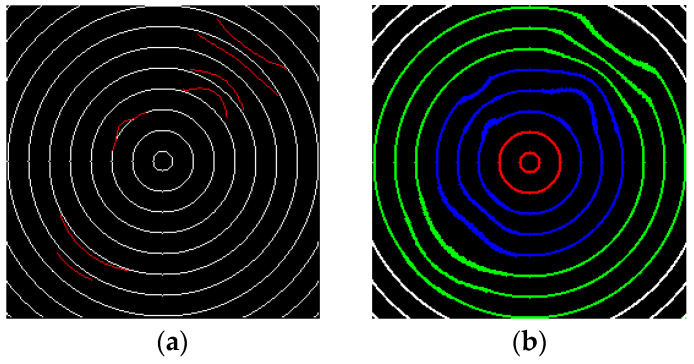
Two resulting deformation images: (**a**) a difference image labeled by detected deformations in red, and (**b**) a difference image labeled according to three distinct deformation severities (severe level in red, average level in blue, and slight level in green).

**Figure 9 sensors-23-04497-f009:**
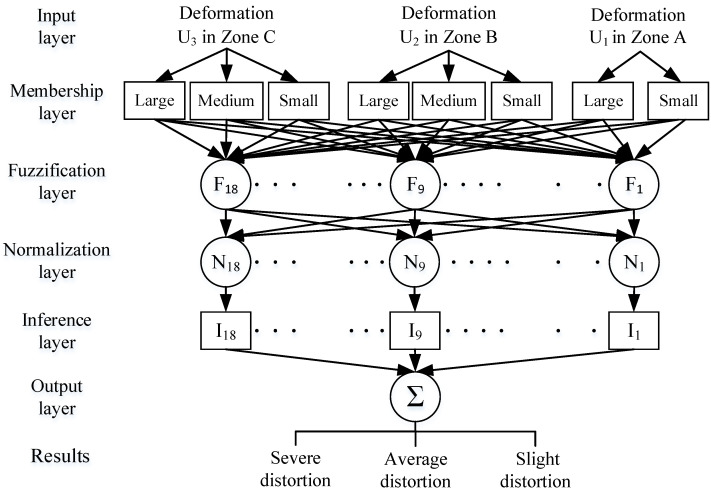
The structure diagram of the proposed ANFIS for determining deformation levels.

**Figure 10 sensors-23-04497-f010:**
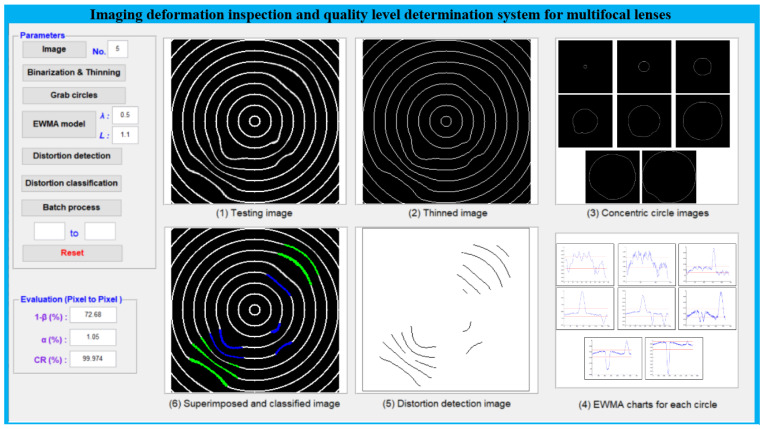
The implemented system with user interface design shows all the processes of the proposed method using the concentric circular pattern.

**Figure 11 sensors-23-04497-f011:**
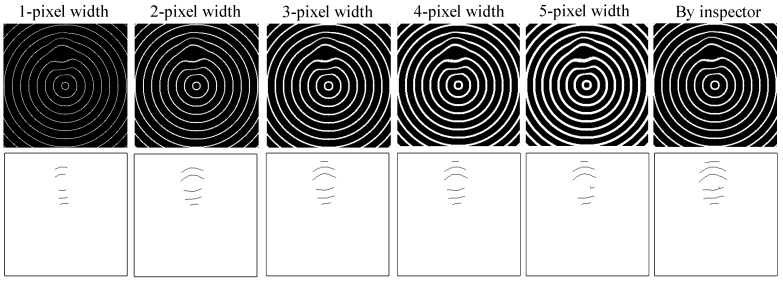
The images captured by the suggested method employing patterns of concentric circles with line widths of five kinds of pixel sizes and the results of a defect sample.

**Figure 12 sensors-23-04497-f012:**
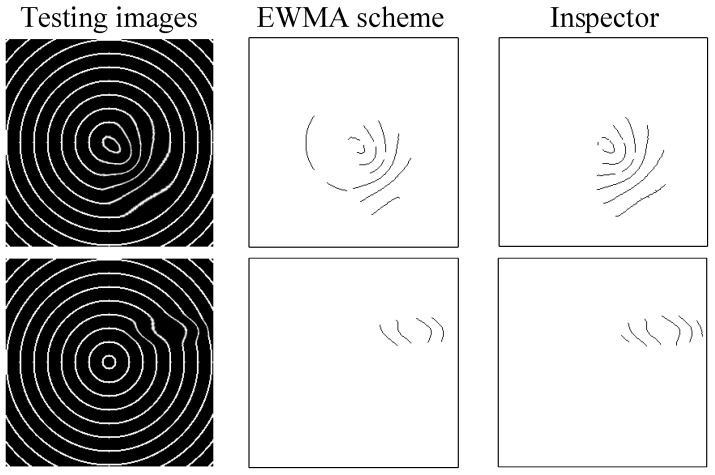
Some outcomes of concentric imaging deformation inspection by employing EWMA slight deviation control scheme.

**Figure 13 sensors-23-04497-f013:**
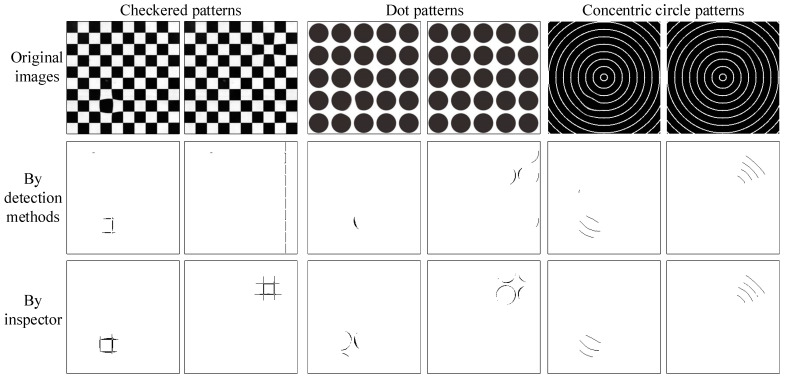
Some results of the proposed method and inspector for imaging deformation inspection employing three conventional norm patterns.

**Figure 14 sensors-23-04497-f014:**
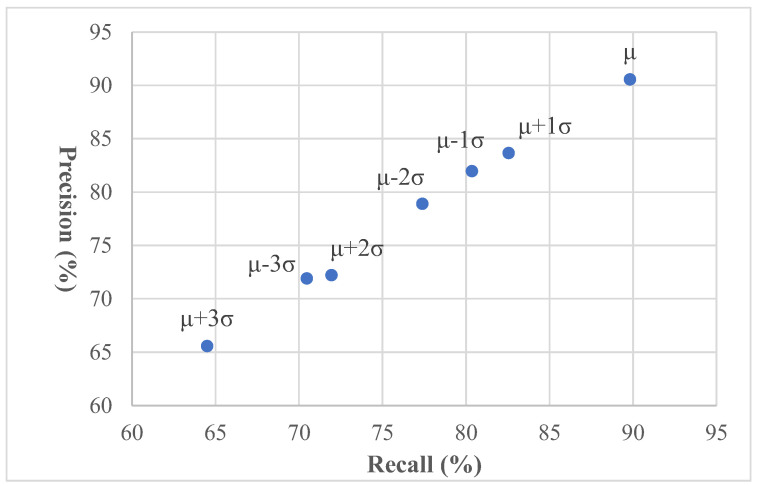
A PR chart of imaging deformation inspection by the proposed method under different lighting conditions.

**Figure 15 sensors-23-04497-f015:**
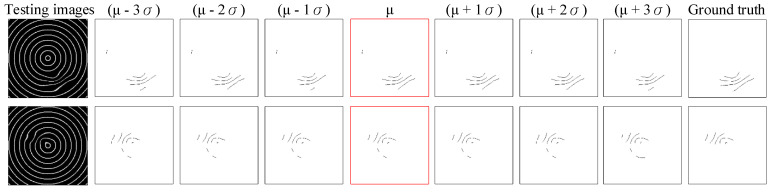
Some detection outcomes of imaging deformation inspection performed by the suggested approach for systematic changes in image lighting (the images with red frames are processed from the selected test image set).

**Table 1 sensors-23-04497-t001:** The input and output items of the suggested FIS model.

	Inputs			Outputs
Features	U_1_: Deformation measure in zone A	U_2_: Deformation measure in zone B	U_3_: Deformation measure in zone C	Y: Distortion levels
Degrees	A_1_: Small A_2_: Large	B_1_: Small B_2_: Medium B_3_: Large	C_1_: Small C_2_: Medium C_3_: Large	Y_1_: Slight Y_2_: Average Y_3_: Severe

**Table 2 sensors-23-04497-t002:** The corresponding membership functions, fuzzy sets, and ranges of the input measures.

Input Items	Membership Functions of Measures	Fuzzy Sets and Ranges of Measures
Deformation measure U_1_ in zone A	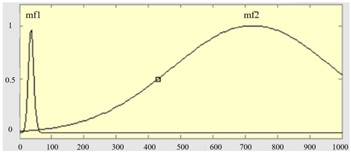	$\mu_{A_{1}}\left(u_{1};\;8.3,\;4.8\right)\;\mu_{A_{2}}\left(u_{1};\;1084,\;765\right)$
Deformation measure U_2_ in zone B	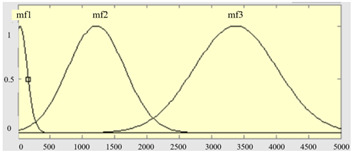	$\mu_{B_{1}}\left(u_{2};\;69,\;17.6\right)\;\mu_{B_{2}}\left(u_{2};\;385,\;895.7\right)\;\mu_{B_{3}}\left(u_{2};\;384,\;2392\right)$
Deformation measure U_3_ in zone C	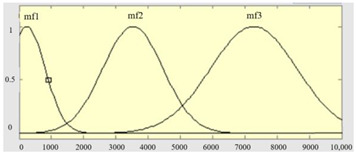	$\mu_{C_{1}}\left(u_{3};287,\;117.2\right)\;\mu_{C_{2}}\left(u_{3};468.5,\;1766\right)\;\mu_{C_{3}}\left(u_{3};670,\;3637\right)$

**Table 3 sensors-23-04497-t003:** Performance metrics for deformation defect detection on captured images by the suggested method employing patterns of concentric circles with line widths of six pixel sizes.

Line Thicknesses	1 Pixel	2 Pixels	3 Pixels	4 Pixels	5 Pixels	6 Pixels
Recall (%)	54.36	80.72	80.12	75.90	79.15	77.49
Precision (%)	94.93	96.02	96.02	90.47	95.32	91.06

**Table 4 sensors-23-04497-t004:** Performance metrics for detecting deformed regions and determining the quality levels of multifocal glasses by the proposed approach.

Deformation Detection Techniques	EWMA Control Scheme		
Recall (%)	81.09		
Precision (%)	89.06		
Processing time (s)	0.2847		
Quality level determination models	BPN	ANFIS	GA based ANFIS
Accuracy (%)	70.00	70.67	94.00

**Table 5 sensors-23-04497-t005:** Performance metrics of imaging deformation detection by the suggested approach employing three conventional norm patterns.

Norm Patterns	Hough Transform-Based Methods [22]		Concentric Circular Pattern
	Checkered Pattern	Dot Pattern	
Recall (%)	33.24	58.20	77.03
Precision (%)	37.64	81.22	76.86
Accuracy (%)	94.70	98.94	99.47

**Table 6 sensors-23-04497-t006:** Performance metrics of imaging deformation inspection using the suggested approach for changing the brightness of the image illumination.

Lighting Intervals	(μ − 3σ)	(μ − 2σ)	(μ − 1σ)	μ	(μ + 1σ)	(μ + 2σ)	(μ + 3σ)
Recall (%)	70.46	77.38	80.35	89.81	82.54	71.93	64.49
Precision (%)	71.92	78.91	81.97	90.58	83.66	72.23	65.57
Accuracy (%)	99.89	99.92	99.92	99.96	99.93	99.89	99.87

## Data Availability

Not applicable.

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
