# Peer review of "Optical Imaging Deformation Inspection and Quality Level Determination of Multifocal Glasses"

_sensors, 2023, doi:10.3390/s23094497_

Round 1

Reviewer 1 Report

The research presented in this paper proposes an optical imaging deformation inspection method for determining the quality level of multifocal glasses. The aim of this research is to replace professional assessors with an automatic deformation defect detection system. The methodology involves using concentric circles and enhancing clarity, followed by detecting the deviation of the centroid radius to identify any deformations. Fuzzy membership functions and inference regulations are then established, and a mixed model is used to determine the quality grade. The results indicate an accuracy rate of 94%.

Although the manuscript is well written, there are several areas where minor corrections could improve its quality.

Firstly, the abstract could benefit from the addition of two sentences, one describing how the research problem is being addressed by the scientific community and any constraints, and the other discussing the significance of the research for the community.

Secondly, the organization of the article should be presented at the end of the introduction section.

Thirdly, the literature review could be improved by providing a tabulated representation of data for better comparison between existing techniques and the proposed method.

Fourthly, a block diagram of the sequence steps (1 to 5) used in the research should be included in the methodology section to aid comprehension.

Fifthly, the image used in Table 2 is blurry, and units are not available on the X-axis and Y-axis.

Finally, the conclusion section is too lengthy and should be made more concise. Some of the content could be moved to the results section or a separate discussion section

Reviewer 2 Report

1. One suggestion is to include more information on the limitations and potential sources of error in the proposed method. While the results are promising, it is important to acknowledge the potential pitfalls and potential sources of error in the proposed method to ensure its accuracy and reliability in practical applications.

2.  It may also be beneficial to explore the use of additional imaging techniques or sensors to enhance the accuracy of the detection system. For example, the use of machine vision or 3D scanning could provide additional data points that may improve the accuracy of the system.

3. The study could benefit from including a more detailed explanation of the fuzzy membership functions and inference regulations used to quantify the severity of the deformation. This would help readers better understand the technical details of the method and how it can be applied in practice.

4. It may be useful to include more information on the computational resources required to run the detection system, such as the processing time and memory requirements. This information could help potential users understand the practical feasibility of implementing the system in their manufacturing processes.

5. The study could benefit from a more in-depth discussion of the practical applications of the proposed method, including potential benefits to consumers and manufacturers. This could help increase the relevance and impact of the research in the field of multifocal glasses.

6. Lastly, the study could benefit from including more data and testing under a wider range of conditions to demonstrate the robustness and effectiveness of the proposed method in various scenarios. This would help increase the reliability and confidence in the system's ability to detect deformation defects in multifocal glasses.

7. One suggestion for improving the writing style is to simplify and streamline the language used in the paper. The technical language and jargon may be difficult for some readers to understand, particularly those without a background in the field of optics or image processing. Simplifying the language can help make the research more accessible and understandable to a broader audience.

8. The paper could benefit from a more structured and organized approach to presenting the research. The current flow of the paper can be difficult to follow at times, particularly when explaining the technical details of the detection system. Reorganizing the information and presenting it in a more logical and coherent manner can help improve the clarity and readability of the paper.

9. It may also be helpful to include more visuals or diagrams to illustrate the technical concepts and processes described in the paper. This can help readers better understand the technical details and improve their comprehension of the proposed method.

10. Lastly, it is important to ensure that the paper is free of errors and inconsistencies. Proofreading the paper carefully and revising any errors or unclear passages can help improve the overall quality of the paper and ensure that it is professional and polished.

Reviewer 3 Report

The paper has a high scientific as well as utilitarian value. Applies to Optical Imaging Deformation Inspection and Quality Level Determination of Multifocal Glasses. To analyze the issues raised, the authors use artificial intelligence methods and fuzzy reasoning methods. The research results mentioned in the abstract as well as in the discussion and conclusions are very promising. Autors used a mixed model incorporating a network-based fuzzy inference and a genetic algorithm is applied to determine a quality grade for the deformation severity of detected defects. Testing outcomes show that the proposed methods attain a 94% accuracy rate of the quality levels for deformation severity, an 81% recall rate of deformation defects, and an 11% false positive rate for multifocal glass detection. I recommend the article to be published in the presented form.

Author Response

Dear Reviewer,

Thank you for your valuable comments and suggestions on our manuscript. We appreciate the opportunity to improve our work based on your feedback.

Point 1: I recommend the article to be published in the presented form.

Response 1: Thanks!

Reviewer 4 Report

Multifocal glasses is a new type of lens that can fit both near-sighted and far-sighted vision on the same lens. The curvature varies of the lens will lead to optical deformations. This manuscript proposes an approach of optical deformation Inspection dedicated for multifocal glasses. The feasibility and performance are demonstrated in a series of experiments.

Comments:

(1) There are multiple procedures in the proposed approach. It is recommended to present a flow chart of the whole approach.

(2) As compared and discussed in Section 4.2, the precision rate of the CUSUM is quite low. It is not necessary to present this method.

(3) Performance of various line thicknesses are compared in Section 4.1. But how can you control the line thicknesses in experiments?

(4) How to determine the standard brightness of the image illumination?

Round 2

Reviewer 2 Report

Paper is Accepted in current form